# The Potential Effect of Lidocaine, Ropivacaine, Levobupivacaine and Morphine on Breast Cancer Pre-Clinical Models: A Systematic Review

**DOI:** 10.3390/ijms23031894

**Published:** 2022-02-08

**Authors:** Ana Catarina Matos, Inês Alexandra Marques, Ana Salomé Pires, Ana Valentim, Ana Margarida Abrantes, Maria Filomena Botelho

**Affiliations:** 1Faculty of Pharmacy, University of Coimbra, 3000-548 Coimbra, Portugal; uc2017246848@student.uc.pt (A.C.M.); ines.marques@student.uc.pt (I.A.M.); 2Coimbra Institute for Clinical and Biomedical Research (iCBR) Area of Environment, Genetics and Oncobiology (CIMAGO), Institute of Biophysics, Faculty of Medicine, University of Coimbra, 3000-548 Coimbra, Portugal; mabrantes@fmed.uc.pt; 3Center for Innovative Biomedicine and Biotechnology (CIBB), University of Coimbra, 3000-548 Coimbra, Portugal; 4Clinical Academic Center of Coimbra (CACC), 3004-561 Coimbra, Portugal; anavalentim@sapo.pt; 5Anaesthesiology Service, Centro Hospitalar e Universitário de Coimbra (CHUC), 3004-561 Coimbra, Portugal

**Keywords:** breast carcinoma, lidocaine, ropivacaine, levobupivacaine, morphine, methadone

## Abstract

Breast cancer (BC) is one of the most common types of cancer and the second leading cause of death in women. Local anaesthetics (LAs) and opioids have been shown to influence cancer progression and metastasis formation in several pre-clinical studies. However, their effects do not seem to promote consensus. A systematic review was conducted using the databases Medline (via PubMed), Scopus, and Web of Science (2010 to December 2021). Search terms included “lidocaine”, “ropivacaine”, “levobupivacaine”, “morphine”, “methadone”, “breast cancer”, “breast carcinoma” and “breast neoplasms” in diverse combinations. The search yielded a total of 784 abstracts for initial review, 23 of which met the inclusion criteria. Here we summarise recent studies on the effect of analgesics and LAs on BC cell lines and animal models and in combination with other treatment regimens. The results suggest that local anaesthetics have anti-tumorigenic properties, hence their clinical application holds therapeutic potential. Regarding morphine, the findings are conflicting, but this opioid appears to be a tumour-promoting agent. Methadone-related results are scarce. Additional research is clearly required to further study the mechanisms underlying the controversial effects of each analgesic or LA to establish the implications upon the outcome and prognosis of BC patients’ treatment.

## 1. Introduction

In 2020, 355,000 women were diagnosed with breast carcinoma (BC) in Europe, constituting 13.3% of all cancer cases detected that year [1]. Globally, this type of cancer was diagnosed in about 2.3 million women, and later that year it was considered the most prevalent type of cancer, reaching 7.8 million women in the past 5 years [2]. Moreover, 685,000 deaths were reported, commonly caused by metastization and tumour recurrence [2,3].

Early diagnosis and adequate treatment provide patients with a greater probability of cure and a better prognosis, since intervention is possible in the earliest stages of the disease and when treatment is, in most cases, more effective. The early diagnostic tests commonly used include clinical breast examination, imaging techniques (mammography, ultrasonography, and magnetic resonance imaging) and in suspected cases, biopsy under local anaesthesia [4].

Currently, surgery, radiotherapy, and systemic treatment, such as chemotherapy, are the three main treatments used in the therapeutic scheme for BC [5], with surgical resection of the primary tumour being the method associated with a better prognosis [3].

The perioperative period has been one of the targets of increased attention among the scientific community, due to its potential influence on the development of metastases and tumour recurrence. Perioperative factors such as pain, stress and anaesthetic techniques are increasingly recognized for affecting the necessary balance for the development of metastases, which combines metastatic immune activity and the ability of cancer cells to metastasize [6].

Despite the treatment advances achieved in the oncology field, during surgery there is immunosuppressive stress induction and cancer cells can be inadvertently disseminated into circulation, increasing the risk of metastases [7,8]. In clinical practice, surgery for BC may occur under general anaesthesia supplemented or not by regional or local anaesthetics (LAs). In particular, amide-linked LAs, such as lidocaine, ropivacaine and levobupivacaine are commonly used for thoracic paravertebral block in BC surgery and to reduce chronic pain. In addition, intravenous lidocaine infusion is increasingly used in multimodal analgesic treatment [9,10]. Studies report that the application of LAs during surgery is associated with lower recurrence and metastasis formation and, in turn, a higher survival rate [9,11]. Regardless of the local anaesthetic applied, they are associated with anti-tumour effects, involving the prevention of cancer cell proliferation, migration, and invasion [9]. As a result, surgical stress response is attenuated, and natural killer (NK) cells activity preserved, thereby contributing to the rejection of tumour cells [3,6]. Similar results shown by Xuan et al. [12], demonstrated that bupivacaine is capable of directly inhibiting prostate and ovarian cancer cell viability, proliferation, and migration at clinically relevant concentrations (i.e., drug concentrations or doses known to be achievable and efficacious in patients) [13]. Moreover, Baptista-Hon et al. [14] demonstrated that ropivacaine inhibits Na_V_1.5 channel function on circulating colon cancer cells, an action that attenuates invasion.

Cancer-related pain is a problem that plagues most patients with BC. The incidence of this pain increases from diagnosis to more advanced stages, and in some cases the pain can be severe and debilitating [15]. The ineffectiveness in controlling cancer-related pain can lead to an exacerbated and prolonged stress response, resulting in immunosuppression and promoting cancer cell spread in the postoperative period [16]. Opioid administration in clinical practice is considered the most effective treatment of cancer-related pain, both perioperative and chronic [3]. Generally, morphine is the opioid of choice for pain management in advanced stages of the disease. It is considered the most effective clinical analgesic available and is mainly used in order to increase patients’ quality of life [6]. On the other hand, methadone, a long-lasting synthetic opioid associated with drug addiction treatment, has been used as an alternative for the treatment of cancer-related pain [17]. Nevertheless, although opioids have strong effects on pain relief, their effect on the development of tumours remains controversial, as there are studies that report a promoting effect and others reporting an inhibitory effect of cancer cells growth, as well as on NK cell activity [6]. Consequently, this arises the need of more preclinical studies to clarify this subject in order to optimize the choice of the best LA and analgesia options for BC patients. 

Moreover, it is crucial to understand whether the concentration in which a drug is applied may influence the effect it exerts on a patient and its implication on the mode-of-action of other drugs administered to them. Clinically recommended concentrations of lidocaine, ropivacaine, levobupivacaine and morphine, as well as their maximum single doses are listed in Table 1. These are valid for adults weighing 70 kg. Regarding morphine, the concentration refers to the management of cancer-related pain and, although it does not have a maximum dose, its minimum lethal dose is 120 mg per dose. Nevertheless, it can range from 60 mg to 3000 mg per dose for susceptible or tolerant patients, respectively [18].

Thus far, there is no literature gathering all the existing information focused on the influence of opioids and LAs on the behaviour of BC tumours and therapeutic response, as well as the molecular mechanisms underlying theses effects, particularly for the LAs ropivacaine, lidocaine and levobupivacaine and the opioids morphine and methadone. In this regard, this review emerges and aims to map and analyse the preclinical studies existing in the literature to assess the applicability of these drugs in BC treatment and its mechanism of action.

## 2. Results

The initial search identified 1027 references, of which 321 were from Medline (Pubmed), 315 from Scopus and 391 from Web of Science. After removing the duplicates, 784 references were reviewed by title and abstracts, resulting in 62 selected articles for full reading. After full-text analysis, 23 references were eligible for analysis and data extraction according to the purposes of this systematic review, as shown in Figure 1.

The data collected from each selected article were gathered in Table 2. The analyses were grouped according to the LA or analgesic in study, namely lidocaine, ropivacaine, levobupivacaine and morphine. Concerning lidocaine, 10 articles were found that were fully integrated in the studied subject, of which 7 corresponded to in vitro studies, 1 in vivo study and 2 with both type of studies. For long-acting anaesthetics, ropivacaine and levobupivacaine only 4 and 3 in vitro studies were found that were related to the subject of BC, respectively. Moreover, 1 article with in vitro and in vivo studies concerning ropivacaine was retrieved from the search. In parallel, a total of 9 articles were found that reported the effect of morphine on the therapeutic response of BC, of which 3 corresponded to in vitro studies, 3 in vivo studies and 3 with both types of studies. Regarding methadone, there are few data since 2010 and those that exist do not fully integrate the topic of BC, so these have been excluded from the table and no details will be provided regarding this opioid. 

Among all the studies analysed, 22 assessed the effect of these drugs on the behaviour of tumour cell lines. Three of these studies, namely Lirk et al. [23], Ge et al. [24] and Gong et al. [25] also examined the combination and influence of these drugs with chemotherapeutic agents. Freeman et al. [26] assessed the combination with general anaesthesia during surgery.

**Figure 1 ijms-23-01894-f001:**
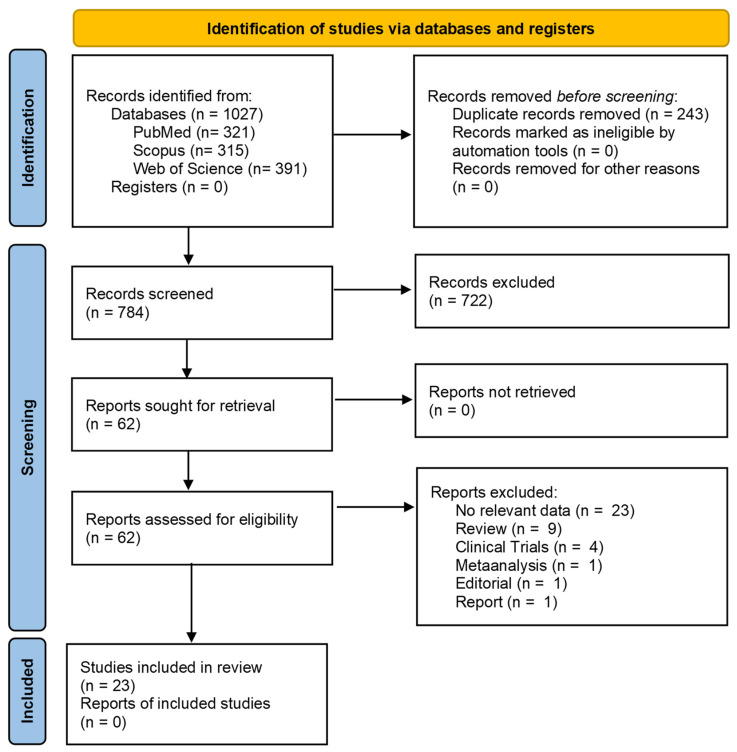
PRISMA flow diagram of the study research methodology in the literature. Adapted from PRISMA Group [27]. * Clinical studies, reviews, meta-analyses, editorials, opinion pieces, and articles not providing the effect of pain medicine on therapeutic response to BC, articles written in languages other than English, written before 2010 or unavailable as complete articles were excluded.

**Table 2 ijms-23-01894-t002:** The effect of lidocaine, morphine, ropivacaine and levobupivacaine on BC cell lines and its influence in other treatment regimens.

Authors Year	Type of Study	Drug Concentration	Outcome	Observations
Lidocaine				
Liu et al. [28] 2021	In vitro MDA-MB-231 AU565 T47D MDA-MB-468 MCF-7 BT474 BT-20	0.3–3 mM	↓	Viability, migration and TRPM7 function on BC cell lines were tested. Lidocaine at 1 and 3 mM (24 h) significantly suppresses the viability of BC cell lines with exception of MCF-7, and with greater effects on AU565, T47D and BT-20 cell lines. Lidocaine at 0.3 mM (24 h) only inhibited the viability of AU565 cell line. Lidocaine at 1 and 3 mM (24 h) suppressed cell migration in all the cell lines but at 0.3 mM the migration is only supressed on MDA-MB-231, AU565, and BT474. TRPM7 plays a role in mediating lidocaine’s effects on viability and migration of MDA-MB-231, AU565, T47D and MDA-MB-468.
Lin et al. [29] 2021	In vitro MCF-7	0.01–0.2 mmol/L	↓	Lidocaine inhibited proliferation, migration, and invasion of BC cell line MCF-7 by modulating the MicroRNA-495-3p/Fibroblast Growth Factor 9 axis. The overexpression of FGF9 inhibited the inhibitory effect of lidocaine on the proliferation, migration, and invasion of breast cancer MCF-7 cells.
Freeman et al. [26] 2019	In vivo Female BALB/c with 4T1 tumour cells	1.5–2 mg/kg/h	↓	Perioperative administration of lidocaine in a BC murine model of surgery, during sevoflurane anaesthesia, reduced the metastatic burden of lung tissue but not the hepatic colonies. No statistical differences were found in serum VEGF and IL-6 concentrations between groups 4 weeks after perioperative administration. Lidocaine infusions were stopped before the postoperative period, presenting a study limitation.
Chamaraux-Tran et al. [30] 2018	In vitro MCF-10A MCF-7 MDA-MB-231 SKBr3 HER^2+^	0.1–10 mM	↓	The viability of MCF-7 and SkBr3 HER^2+^ cell lines reduced significantly at 1 mM lidocaine and more (4 h). The MDA-MB-231 and MCF10A cell lines were more sensitive to lidocaine 4 h treatment, showing a significant viability reduction starting at 0.5 mM. Exposure to lidocaine at 0.1 mM (24 h) resulted in a marked inhibition of the migration of both MDA-MB-231 and SkBr3 HER^2+^ cell lines compared to MCF10A normal cells. MCF-7 and MCF-10A cell lines did not shown any significant migratory ability after lidocaine treatment. Lidocaine (0.1 mM) compromised the anchorage-independent growth of the MDA-MB-231 cell line.
	In vivo SCID female mice inoculated with MDA-MB-231 cells	100 mg/kg	↓	Intraperitoneal lidocaine improved survival of mice model of MDA-MB-231 peritoneal carcinomatosis and reduced tumour growth.
Li et al. [31] 2018	In vitro MCF-10A	10–100 μM	=	Lidocaine decreased viability and induced significant cellular toxicity only on tumour cells (MDA-MB-231 and MCF-7 cells) exclusively with concentrations range of 0.3 to 10 mM (48 h). Also, a significant apoptotic response was observed, in this range, for MDA-MD-231 cells. At plasma concentrations (10 μM), lidocaine (24 h) promoted cell cycle arrest from phase S to phase G2/M in MDA-MB-231 cell line. Interestingly, at 10× plasma concentration (100 μM) the shift from G0/1 to S phase was already seen after 6 h. A selective effect was shown considering that the viability of non-cancer human breast epithelial MCF10A cells was not affected.
		0.3–10 mM	=	
	MCF-7	10–100 μM	=	
		0.3–10 mM	↓	
	MDA-MB-231	10–100 μM	=	
		0.3–10 mM	↓	
Agostino et al. [32] 2018	In vitro MDA-MB-231	0.001–100 μM	↓	Lidocaine at concentrations of 10 or 100 μM (24 h) inhibited CXCR4 and mediated migration of MDA-MB-231 cell line, involving changes in intracellular calcium release and cytoskeleton remodelling.
Jiang et al. [33] 2016	In vitro MDA-MB-231	0.01–0.1 μM	=	Inhibitory effect on cell invasion was enhanced with increasing concentrations of lidocaine. With 10 μM, 100 μM and 1 mM of lidocaine (24 h) cell migration of the MDA-MB-231 cells was remarkably inhibited. Lidocaine effects on migration and invasion could occur partly as a result of the downregulation of TRPV6 expression. Lidocaine was able to significantly decrease cell viability in a concentration-dependent manner from 1 to 10 mM (4 h). However, lower concentrations (≤1 mM) of lidocaine exhibited no marked cytotoxicity.
		1–10 mM	↓	
Chang et al. [34] 2014	In vitro MCF-10A	2–32 mM	=	Lidocaine decreased MCF-7 cells viability, increasing cell death by apoptosis, in a dose- and time-dependent manner (24 h). In vitro apoptotic effects of lidocaine are reproducible in vivo.
	MCF-7	1–16 mM	↓	
	In vivo Female BALB/c nude mice inoculated with MCF-7 cells	21.3 mM	↓	
Lirk et al. [23] 2014	In vitro BT-20 MCF-7	10–100 μM	↓	Authors incubated BC cell lines with lidocaine to assess demethylating properties. Lidocaine (10 or 100 μM) and the chemotherapeutic 5-aza-2′-deoxycytidine (DAC) at 0.1 or 0.5 μM demonstrated to have additive demethylating effects in BT-20 cell line after 72 h treatment. In MCF-7 cells, only the combined treatment with 0.5 μM DAC and 10 μM lidocaine revealed a stronger demethylation. The concentrations used were insufficient to cause direct cytotoxicity. Methylation bases between the cell lines shown to have different properties. Biological heterogeneity may have had a role in different outcomes of anaesthetics interventions.
Lirk et al. [35] 2012	In vitro BT-20 MCF-7	0.01–0.1 mM	=	Treatment with 1 mM lidocaine (72 and 96 h) resulted in significant reductions in cell number, while lower concentrations of local anaesthetics had no effect. There was an increase in the apoptosis rate upon lidocaine (1, 0.1, and 0.01 mM) 72 and 96 h treatment. At clinically relevant concentrations (1 mM), lidocaine demethylated DNA of MCF-7 cells after 72 h. Whereas treatment with 0.1 and 0.01 mM lidocaine revealed a significant demethylation after 72 and 96 h. In BT-20 cell line, was observed a dose-dependent decrease in DNA methylation in response to lidocaine (1, 0.1, and 0.01 mM) after 72 and 96 h. Demethylating tumour-suppressive effects may only be detectable in specific types of cancer due to differential methylation profiles. The cell lines used may have had genotypic and phenotypic derivations since their validation.
		1 mM	↓	
Ropivacaine				
Zhao et al. [36] 2021	In vitro MDA-MB-231 MDA-MB-468 MCF-10A SKBr3 HER^2+^ MCF7 BT474	1 mmol/L	↓	Ropivacaine inhibited proliferation, decreased migration and invasion and induced apoptosis of breast cancer cells MDA-MB-231 and MCF-7. This LA might inhibit the progression of all the BC cell lines tested by modulating the miR-27b-3p /YAP axis.
	In vivo Balb/c nude mice injected with MDA-MB-231 cells	40 μmol/Kg	↓	Treatment with ropivacaine repressed the cell growth of MDA-MB-231 cells in vivo, while miR-27b-3p inhibitor could reverse this effect. Thus, confirming the results obtained in vitro.
Castelli et al. [37] 2019	In vitro MDA-MB-231	5–1000 μM	↓	Ropivacaine at 5 μM and more decreased significantly cell viability after 48 h of treatment. This LA (50 μM) resulted in 50% mortality of MDA-MB-231 cell line after 24 h treatment. Ropivacaine promoted apoptosis paralleled by the inactivation of survival pathways, such as PI3K/Akt/GS3K/β-catenin. This LA was able decrease cell proliferation by inactivating Wnt/GSK3β/β-catenin pathway. Ropivacaine was able to decrease RhoA and the active form of FAK protein level, indicating a reduction in cell invasion and migration.
Li et al. [31] 2018	In vitro MCF-10A	3.5–35 μM	=	Ropivacaine decreased viability, inhibited migration, and induced significant cellular toxicity of MDA-MB-231 only in concentrations of 0.3 to 10 mM (48 h). At plasma concentrations (3.5 μM), ropivacaine (24 h) promoted cell cycle arrest from phase S to phase G2/M in MDA-MB-231 cell line. Curiously, 24 h treatment with this LA blocked cell cycle before mitosis of MDA-MB-231 cells treated at 10× plasma concentrations (35 μM). Ropivacaine did not affect viability or cellular toxicity of non-tumorigenic human breast epithelial MCF10A cells.
		0.3–10 mM	=	
	MCF-7	3.5–35 μM	=	
		0.3–10 mM	↓	
	MDA-MB-231	3.5–35 μM	↓	
		0.3–10 mM	↓	
Gong et al. [25] 2018	In vitro MDA-MB-468 SKBr3 HER^2+^	0.1–1 mM	↓	After 72 h of treatment, ropivacaine at concentrations of 0.5 and 1 mM significantly inhibited proliferation and induced apoptosis in a concentration-dependent manner. SKBr3 HER^2+^ cells appear to be more sensitive to ropivacaine than MDA- MB-468 cells. Ropivacaine significantly inhibited growth, survival, and anchorage-independent colony formation (72 h). Interestingly, ropivacaine at 0.5 mM significantly inhibits colony formation but does not affect growth and survival. Ropivacaine inhibited mitochondrial respiration by suppressing mitochondrial respiratory complex I and II activities, leading to energy depletion, oxidative stress, and damage. It was demonstrated a synergism between ropivacaine and 5-FU, likely by suppressing Akt/mTOR signalling pathway.
Lirk et al. [23] 2014	In vitro BT-20 MCF-7	3–30 μM	=	Ropivacaine showed no cytotoxic effect in either BC cell line. Ropivacaine after a 72 h treatment decreased methylation in BT-20 cells. Ropivacaine plus DAC revealed no increased demethylating effect in BT-20 or MCF-7 cells. Methylation bases between the cell lines shown to have different properties. Biological heterogeneity may have had a role in different outcomes of anaesthetic interventions.
Levobupivacaine				
Kwakye et al. [38] 2020	In vitro MCF-7 MDA-MB-231	1–3 mM	↓	Levobupivacaine inhibited proliferation and promoted apoptosis in BC cells. Levobupivacaine after a 24 h treatment significantly decreased in the invasion ability of MCF-7 and MDA-MB-231 cells in a dose-dependent manner. Findings demonstrated a significantly increase of BAX expression and were associated with a decreased of BCL-2 expression and inhibition of PI3K/Akt/mTOR signalling pathway.
Castelli et al. [37] 2019	In vitro MDA-MB-231	5–1000 μM	↓	Levobupivacaine at 10 μM and more decreased significantly cell viability after 24 h of treatment. This LA (50 μM) resulted in 50% mortality of MDA-MB-231 cell line after 24 h treatment. Levobupivacaine promoted the inactivation of survival pathways such as PI3K/Akt/GS3K/β-catenin, contributing to cell death by apoptosis. Levobupivacaine was able decrease cell proliferation by inactivating Wnt/GSK3β/β-catenin pathway. This LA was able to decrease RhoA and the active form of FAK protein level, indicating a reduction in cell invasion and migration.
Li et al. [31] 2018	In vitro MCF-10A	2.5–25 μM	=	Levobupivacaine decreased viability, significantly inhibited migration, and induced significant cellular toxicity of MDA-MB-231 and MCF-7 cells only in concentrations of 0.3 to 10 mM (48 h). Levobupivacaine at plasma concentrations (2.5 μM) promoted a cell cycle arrest from phase S to phase G2/M in MDA-MB-231 cell line (24 h). Interestingly, at 10× plasma concentration (25 μM) the shift from G0/1 to S phase was already seen after 6 h. Levobupivacaine did not affect viability or cellular toxicity of non-tumorigenic human breast epithelial MCF10A cells.
		0.3–10 mM	=	
	MCF-7	2.5–25 μM	=	
		0.3–10 mM	↓	
	MDA-MB-231	2.5–25 μM	=	
		0.3–10 mM	↓	
Morphine				
Cheng et al. [39] 2019	In vitro MDA-MB-231	10 μmol/mL	↑	Morphine promoted lung metastasis 3 weeks after BC surgery in animal models. Morphine promoted postoperative recurrence, tumour proliferation and angiogenesis and reduced tumour cell apoptosis. PI3K-c-Myc signalling pathway may be related to angiogenesis promoted by morphine. Authors did not describe how morphine promoted the unexpected increased expression of TSP-1.
	In vivo BALB/c-nu specific-pathogen-free mice with MDA-MB-231 cells	10 mg/kg	↑	
Chen et al. [40] 2017	In vitro MCF-7	0.01–10 μM	↓	Morphine inhibited cell growth by blocking the cell cycle and promoted apoptosis in MCF-7 cells. Naloxone could not reverse morphine effects, which indicated that the inhibition of cell growth and proliferation by morphine could be an independent effect, not associated with opioid receptors.
Bimonte et al. [41] 2015	In vitro MCF-7 MDA-MB-231	1–100 μM	↑	Morphine enhanced proliferation, migration, and inhibited apoptosis of BC cell lines at 48 h in a dose-dependent manner.
	In vivo Foxn1^nu/nu^ mice with MDA-MB-231	0.714–1.43 mg/kg/day	↑	Morphine promoted tumour growth and angiogenesis.
Doornebal et al. [42] 2015	In vivo Female wild-type syngeneic FVB/N mice	10 mg/kg/12 h	=	Morphine in the presence or in the absence of surgery-induced tissue damage and pain, neither facilitated de novo metastatic dissemination nor promoted outgrowth of minimal residual disease after surgery. It did not exclude the possibility that anaesthetic techniques may influence the progression of the disease due to the intrinsic properties of the drugs.
	Female MMTV-NeuT mice (BALB/c background)	20 mg/kg/12 h	=	
Niu et al. [43] 2015	In vitro MCF-7 BT549 MCF-10A	1–10 μM	↑	Morphine contributed to chemoresistance via expanding the population of cancer stem cells and promoted tumour growth in vitro. Compared with the normal saline group, morphine group showed a larger tumour volume after 21 days. Morphine enhanced the tumorigenicity of BC cells in vivo, however, this effect could be blocked by nalmefene.
	In vivo NOD/SCID mouse model inoculated with BT549 cells	5–15 mg/kg	↑	
Nguyen et al. [44] 2014	In vivo C3TAG mice henceforth	0.5–1.5 mg/kg/day	↑	Morphine did not affect the onset of tumour development, but it promoted growth of existing tumours, and reduced overall survival in mice. Mast cell activation by morphine might have contributed to increased cytokine and substance P levels, leading to cancer progression and refractory pain.
Ge et al. [24] 2014	In vitro MCF-7	50–1250 μM	↓	Morphine at 250 μM and 1250 μM (48 h) significantly inhibited proliferation and induced apoptosis in MCF-7 cells. In combination with 500 μM of the chemotherapeutic agent 5-Fluorouracil (5-FU) there was an inhibition of proliferation and apoptotic promotion in MCF-7 cells.
Ecimovic et al. [45] 2011	In vitro MDA-MB-231 MCF-7	10–100 ng/mL	↑	Morphine increased both expression of NET1 and cell migration, but not when NET1 was silenced, suggesting that NET1 contributes to directly mediating the effect of morphine on BC cell migration.
Ustun et al. [46] 2010	In vivo BALB/c bearing Ehrlich carcinoma	0.714 mg/kg/day	↑	Morphine was able to induce angiogenesis. The study was not designed to study the underlying mechanism.

↑: enhance cancer; ↓: inhibit cancer grow or metastasis; =: no effect on cancer.

## 3. Discussion

A multidisciplinary approach for the treatment and diagnosis of BC might increase the patients’ survival and quality of life [5]. Despite the curative therapeutic intention in choosing the best-fitting therapy, there are other perioperative factors that might influence the outcome of BC patients [6]. Among others, the use of LAs and opioids were pointed out through a large amount of experimental evidence as part of these factors. Stress, both surgical or BC-related pain, seems to promote cancer cell dissemination and metastases, and simultaneously a decrease in activity of NK cells [47]. On the other hand, LA and opioids were highlighted for their potential effects on pain and stress and thereby on the recurrence of BC and metastases. Thus, the purpose of this review was to identify the studies focused on the potential effects of LAs and analgesics on BC cell lines and animal models and their impact on other treatment regimens, in order to clarify their role and relevance for the treatment of BC patients. To meet this goal, 21 studies were included in this review (Figure 1). Transversally, the effects of LA or analgesics as inhibitors or inducers of tumour proliferation is neither consensual between authors nor for all medicines analysed. In addition, the effect of these drugs in combination with specialised treatments, such as with chemotherapeutic agents or under general anaesthesia, must be highlighted and prioritised in upcoming studies. Regarding methadone, there is a requirement for more studies to be conducted in the future. Therefore, there is a need for further studies on the influence of methadone on the therapeutic response of BC. Generally, most of the studies analysed in this review are focused on lidocaine and morphine. However, studies focused on lidocaine, ropivacaine and levobupivacaine present more consistent results, while more studies are needed on the effects of morphine.

### 3.1. Lidocaine

Lidocaine is an amide-linked LA associated with reduced pain scores and thus reduced need to resort on analgesics [47]. Nevertheless, recent studies have focused on its anti-tumour properties and the surrounding mechanism (Figure 2).

Overall effects of lidocaine on BC cell lines, at concentrations clinically relevant, seem to be promoted in a dose-dependent manner. In doses ranging from 0.3 to 10 mM, it promoted inhibition of viability, invasion, migration, and proliferation selectively depending on the sensitivity of each cell line to this LA [28,29,30,31]. Moreover, lidocaine effects were shown to be selective as all studies made in the non-tumorigenic cell line MCF10A revealed no adverse effects. The study of lidocaine in animal models also showed that its use suppressed metastatic disease and improved animal overall survival regardless of the route of administration [28,30].

The role of lidocaine in supressing migration was reported for several cell lines, however, they remain controversial for MCF-7 cells at ranges of 0.1–10 mM. Lidocaine used at an anti-arrhythmic concentration (10 μM) does not appear to have any major effect on BC cell lines [31]. Nevertheless, MDA-MB-231 cells seem to be the most sensitive to this LA as reported by Agostino et al. [32] and Jiang et al. [33] who demonstrated that 10 or 100 µM lidocaine reduces the migration of this cell line. Moreover, the intraperitoneal administration of a clinically relevant dose of lidocaine (100 mg/kg) on an SCID female mice model inoculated with MDA-MB-231 cells, resulted in tumour growth decrease, without observed adverse effects, confirming the sensitivity of BC tumour to this LA [30]. A significant decrease of the metastatic burden of lung tissue was observed in a metastatic model of BC female BALB/c mice with 4T1 tumour cells, after lidocaine administration, in a bolus of 1.5 mg/kg, followed by an infusion of 2 mg·kg^−1^·h^−1^, during sevoflurane anaesthesia [26]. However, this effect was neither observed for hepatic metastasis nor associated with changes in VEGF and IL-6 concentrations as it was expected by the authors, since tumour cell invasion and spread are typically associated with changes in the cellular and cytokine milieu [26].

The cytotoxic effects observed in vitro were associated with cell death by apoptosis for MDA-MB-231 [31] and MCF-7 [34] cells at ranges of 0.3–16 mM. Particularly, Chang et al. [34] suggested that lidocaine in ranges of 1 to 16 mM inhibits MCF-7 cell growth and viability suggesting that the increased cytotoxicity may be due to cellular apoptosis through activation of extrinsic and intrinsic (mitochondrial) caspase-dependent pathways, in a dose- and time-dependent manner. In vivo studies developed by Chang et al. [34], using a female BALB/c mice inoculated with MCF-7 cells, showed that peritumoral injections of 21.3 mM of lidocaine induced apoptosis, supported with the presence of higher expression of cleaved caspase 7 and a multitude of DNA strand breaks [34]. Moreover, Li et al. [31] associated cell death by apoptosis of MDA-MB-231 cells with a cell cycle arrest in phase S at plasma concentrations (10 μM) after a 24 h treatment. Interestingly, at 10× plasma concentration (100 μM) the shift from G0/G1 to S phase was already seen after 6 h.

Several molecular mechanisms have been proposed for the cytotoxic effects of lidocaine in cancer cells. It was reported that lidocaine effects are mediated by membrane ionic channels, such as melastatin-like transient receptor potential 7 (TRPM7) ion channels. Liu et al. [28] used seven different BC cell lines expressing TRPM7 to evaluate the effects of lidocaine in these channels and its possible role mediating cell migration and proliferation. It was confirmed that exposure to lidocaine (1 and 3 mM) suppresses TRPM7 function associated with a decrease in cell migration and viability of MDA-MB-231, AU565, T47D and MDA-MB-468 cell lines. Still, the knockout of *TRPM7* gene did not lead to a significant suppression of migration and viability probably because there might be more mechanisms involved in the response to lidocaine when a chronic lack of TRPM7 function is present. This effect did not apply to MCF-7 cells [28]. Nevertheless, at a concentration range of 0.1–0.2 mM, Lin et al. [29] showed this LA inhibited MCF-7 cell line proliferation, migration, and invasion by inhibiting the expression of Fibroblast Growth Factor 9 (FGF9) by up-regulating the expression of miR-495-3p, a small single-stranded non-coding RNA shown to supress migration of other types of cancer.

There was also evidence that transient receptor potential cation channel subfamily V member 6 (TRPV6) has a major impact on cellular calcium influx, essential for the migration of tumour cells. Still, the exact role of TRPV6 in tumour progression and development remains unclear. In a TRPV6-expressing cancer cell line, MDA-MB-231, Jiang et al. [33] demonstrated, through the transwell and the wound healing assay, that lidocaine at concentrations from 10 µM to 1 mM inhibits cell invasion and migration. This inhibitory effect was associated with a reduced rate of calcium influx partly caused by TRPV6 downregulation, leading to decreased percentage of cells in the S-phase of the cell cycle [33]. Particularly, the decrease in calcium influx was most prominent at the concentration of 100 µM. Conversely, Agostino et al. [32] using the scratch wound assay and chemotaxis experiments at concentrations from 0.001 to 100 µM, found that lidocaine inhibited MDA-MB-231 cell migration through the inhibition of chemokine CXCL12 and the activity of its receptor CXCR4, whose overexpression has been strongly associated with the metastatic potential of BC cells. Further studies indicated that the suppression of CXCL12-induced CXCR4 signalling leads to the blockage of intracellular Ca^2+^ increase and impairs the cascade of cytoskeleton remodelling, essential to promote polarisation and cell motility during cell migration [32].

The modification of epigenetic information is recognized as a major hallmark of cancer. Still, the magnitude of lidocaine’s effect on BC cell lines may be dependent upon epigenetic features of the tumour type. In oestrogen receptor (ER)-negative, BT-20 cell line, and (ER)-positive MCF-7 cell line, 0.01 to 1 mM of lidocaine demethylated DNA in a time- and dose-dependent manner, suggesting that this LA could regulate epigenetic events [25,35]. In fact, this demethylation of DNA could lead to a reactivation of tumour suppressors and consequently result in tumour growth inhibition [23]. Nevertheless, this effect was much more pronounced in the BT-20 cell line, which is associated with high methylation levels at baseline. Furthermore, when combined with the chemotherapeutic agent DAC, lidocaine had additive demethylating effects on cells [23]. 

Taken together, these results are clinically encouraging and strongly suggest lidocaine as an extremely useful adjunct to BC treatment, not only to reduce tumour growth and metastasis but also to promote apoptosis of existing tumour cells.

### 3.2. Ropivacaine

Ropivacaine is a long-acting amide-linked LA, voltage-sensitive sodium channel inhibitor, commonly used in BC surgery as a perioperative neuraxial anaesthetic and to reduce postoperative pain [25]. 

The overall effects of ropivacaine on BC cells seem to be promoted in a concentration-dependent manner after 48 h of treatment [25,31,37]. In fact, in doses ranging from 0.05 to 10 mM, it promoted cell death and selectively inhibited cell proliferation and migration [27,31,36]. In turn, in doses ranging between plasma concentrations and 10× plasma concentrations (i.e., 3.5 to 35 μM) ropivacaine showed no cytotoxic effect in MDA-MB-231 and MCF-7 cell lines [31]. Similar results were obtained in BT-20 cell line, at concentrations from 3 to 30 μM [23]. In non-tumour cells MCF-10A, ropivacaine appears to have no effect, indicating that this LA only affects cancer cells [31]. The possible mechanisms underlying ropivacaine’s anti-cancer effects are shown in Figure 3.

The cytotoxic effects observed in vitro were associated with cell death by apoptosis for MDA-MB-231 cells with a cell cycle arrest in S phase, after a 24 h treatment with plasma and 10× plasma concentrations of ropivacaine [31]. 

Several molecular mechanisms have been proposed for the cytotoxic effects of ropivacaine in cancer cells. It was reported that ropivacaine effects are supported by the inactivation of the PI3K/Akt/GS3K/β-catenin survival pathway and Wnt/GSK3β/β-catenin cell proliferation pathway, paralleled with an increase of P53 protein and interaction with the inhibitor of cyclin-dependent kinases P21, which leads to a well-established cell cycle arrest after a 24 h treatment with 50 μM of this LA in MDA-MB-231 cells, as shown by Castelli et al. [37]. 

There was also evidence of an association between the Akt/mTOR signalling pathway and mitochondrial function in BC. Gong et al. [25] used MDA-MB-468 and SKBr3 HER^2+^ cell lines to evaluate the effects of ropivacaine on mitochondrial function and its possible role mediating cell apoptosis and proliferation. In fact, a 72 h treatment with 0.5 to 1 mM ropivacaine led to inhibition of cell proliferation and induction of apoptosis of either cell lines in a concentration-dependent manner [25]. Still, the SKBr3 HER^2+^ cell line seemed to be more sensitive to ropivacaine than MDA-MB-438 cells. Furthermore, at this range of concentrations and timeline, this LA supressed the Akt/mTOR signalling pathway, which leads to impaired mitochondrial function and subsequently induced oxidative stress on BC cell lines [25]. Hence, this crosstalk mechanism might play a critical role and should be assessed in future studies. 

Alternatively, it was suggested by Zhao et al. [37] that 1 mM of ropivacaine might induce an inhibitory effect on BC progression by regulating the microRNA-27b-3p/YAP axis. Yes-associated protein (YAP) is a transcriptional co-activator that is abnormally expressed in diverse cancer models, and it has been shown that this LA is able to up-regulate the expression of miR-27b-3p which represses YAP expression and results in BC proliferation suppression and apoptosis stimulation. Similar results were obtained in vivo [37]. Nonetheless, in this study the overexpression of YAP and the miR-27b-3p inhibitor did not fully rescue ropivacaine-reduced BC cell viability [37], suggesting that ropivacaine’s effect on BC cells progression might occur by other additional mechanisms, pointing out the need for future investigation.

Ropivacaine’s effect on BC cell lines may be dependent on epigenetic characteristics of the tumour. In fact, Lirk et al. [23] showed that after a 72 h treatment with this LA, there was a decreased methylation in BT-20 cells. However, this effect was absent in the MCF-7 cell line.

Chemotherapy still represents one of the main treatments used in the therapeutic scheme for BC [5]. Nevertheless, inhibiting tumour growth and mitigating metastasis, along with reducing adverse effects present further challenges that need to be overcome. Therefore, combinational therapy may provide an alternative solution for these challenges. Gong et al. [25] studied the combining effect of ropivacaine with the chemotherapeutic agent, 5-FU in the MDA-MB-468 cell line. Ropivacaine showed to act synergistically with 5-FU inhibiting multiple biological activities of BC, likely by suppressing the Akt/mTOR signalling pathway, suggesting that this LA could be a useful addition to BC treatment [25]. Nonetheless, when combined with the chemotherapeutic agent DAC, ropivacaine revealed no increased demethylating effect in BT-20 or MCF-7 cells [24]. Nevertheless, no studies using animal models were found and therefore future in vivo studies regarding the effect of ropivacaine on breast cancer are recommended.

### 3.3. Levobupivacaine

Levobupivacaine is the long-lasting S-enantiomer of amide-linked bupivacaine, presenting additional advantages, namely its lower toxicity, which makes it safer to use than bupivacaine [48] in nerve blocks, epidural and intrathecal anaesthesia [37].

The results of the evaluation of levobupivacaine use in BC are consistent. Levobupivacaine at 2.5 μM had no major effect on BC cells lines, while in doses from 0.3 to 10 mM it was the most effective LA, comparative to lidocaine, mepivacaine, ropivacaine, bupivacaine, and chloroprocaine, to promote cell death and selectively inhibit cell migration in a concentration-relevant manner [31,38]. In non-tumorigenic cell line, MCF10A, levobupivacaine seemed to have no effect [31], supporting the hypothesis that the mechanism of this LA is selective for cancer cells.

Numerous mechanisms have been associated with the decrease of cell proliferation, induced by this LA (Figure 4). According to Li et al. [31], a 24 h treatment with levobupivacaine at plasma concentrations (2.5 μM) promoted this effect due to a cell cycle arrest in the S phase. Moreover, at 10× plasma concentration (25 μM) the shift from G0/G1 to S phase was also seen after 6 h [31]. It was then suggested that the pronounced cell cycle arrest was associated with inactivation of the PI3K/Akt/GS3K/β-catenin survival pathway and Wnt/GSK3β/β-catenin cell proliferation pathway, paralleled with the increase of P53 and by interacting with the inhibitor of cyclin-dependent kinases P21 [37]. At higher concentrations (1 to 3 mM), Kwakye et al. [38] showed that the inhibition of BC proliferation by levobupivacaine may also be associated with PI3K/Akt/mTOR signalling pathway suppression.

The role of levobupivacaine in inducing cell death by apoptosis have been reported for MDA-MB-231 and MCF-7 cell lines. According to Castelli et al. [37] treatment with 50 μM levobupivacaine resulted in 50% mortality of the MDA-MB-231 cell line after a 24 h treatment. Furthermore, this LA promoted the inactivation of survival pathways such as PI3K/Akt/GS3K/β-catenin, contributing to cell death by apoptosis [37]. This finding is supported by a significant increase of BAX expression and a decrease of BCL-2 expression [38]. Nevertheless, Kwakye et al. [38] associated cell apoptosis with an observed inhibition of the PI3K/Akt/mTOR signalling pathway (Figure 4).

Taken together, this complex crosstalk between the PI3K/Akt/mTOR signalling pathway and PI3K/Akt/GS3K/β-catenin survival pathway might constitute a potential target for levobupivacaine applicability as a cytotoxic agent.

However, despite consistent searching, few in vitro studies were found, and no in vivo studies were retrieved from search. Thus, further studies are recommended on the effect of this LA on the therapeutic response of BC and its adjacent mechanism.

### 3.4. Morphine

There are indications in the literature that morphine can modulate cell migration and adhesion in BC, influencing metastatic potential, however its effect remains controversial [41], possibly due to differences in the experimental design. The possible mechanisms underlying morphine’s effect on BC cells are presented in Figure 5.

In fact, the majority of in vitro studies showed that morphine might increase cell proliferation and migration and inhibit apoptosis at ranges from 10–100 ng/mL and 1 to 100 μM [41,43,45]. Nevertheless, the sensitivity of each cell line to morphine was very different, without a clear dose-response relationship [45]. In contrast, Ge et al. [24] argued that morphine in concentrations ranging from 50–1250 μM has antiproliferative effects in MCF-7 cells. This hypothesis is also supported by Chen et al. [40] who demonstrated that, for the same cell line, morphine in doses ranging from 0.01 to 10 μM, inhibited cell growth by arresting cell cycle progression from G1 to S phase. However, the huge discrepancies in experimental conditions among the studies considered, such as the duration of opioid exposure and the cell density, might justify these discrepancies between effects.

Comparing studies using a similar density per well (from 2 to 5 × 10^3^ cells/well) it is possible to see that antiproliferative effects of morphine on BC cell lines seem to be associated with a reduced exposure time (up to 3 days) [26,39], while a stimulation of proliferative effect appears to be markedly significant only at exposures longer than 3 days [41,43]. Furthermore, when the exposure duration to the opioid does not exceed the 3 days, the effect on BC proliferation seems to alter accordingly to the experimental density utilized per well. In in vitro studies testing lower densities per well (from 2 to 5 × 10^3^ cells/well), a less marked proliferative effect was observed [40,41,43] in comparison to the study of Ecimovic et al. [44] where the experimental density used per well was higher (5 × 10^4^ cells/well).

Alternatively, at experimental concentrations far higher (10 μmol/mL), Cheng et al. [39] proposed that this opioid promotes the metastasis and proliferation of MDA-MB-231 dormant cells through activation of the PI3K-c-Myc signalling pathway. When validated in vivo using a BALB/c-nu/nu specific-pathogen-free inoculated with MDA-MB-231 cells, Cheng et al. [39] showed that morphine promoted postoperative recurrence, tumour proliferation and angiogenesis, as well as a reduction in tumour cell apoptosis [39].

Morphine is extensively used for anaesthetic pre-medication and management of cancer pain in patients undergoing cancer treatment, such as chemotherapy. Thus, understanding how morphine interacts with chemotherapeutic agents may be relevant to improve therapeutic management. Ge et al. [25] argued that morphine in concentrations ranging from 50–1250 μM, in combination with the chemotherapeutic agent 5-FU, inhibited MCF-7 cells proliferation and promoted apoptosis. In turn, Niu et al. [43] used three different BC cell lines to address whether morphine participates in regulation of cancer stem cell properties, which were closely correlated with the chemoresistance of cancer cells and tumour malignancy. It was confirmed that exposure to morphine (1–10 μM) enriched cancer stem cell populations and contributed to the development of chemoresistance and tumour growth [43]. Moreover, in vivo, Niu et al. [42] showed that this opioid at a range from 5 to 15 mg/kg promotes tumorigenesis in an NOD/SCID mouse model. Nevertheless, morphine’s effects in vivo were shown to be discrepant, which according to Ustun et al. [46], might be due to the inhibition of angiogenesis via excessive doses of morphine owing to its non-specific cytotoxicity.

In fact, at clinically recommended doses (0.714 mg/kg), Ustun et al. [46] demonstrated that morphine stimulates angiogenesis and increases BC progression. Similar results were obtained by Bimonte et al. [41] in a heterotopic mouse model of human triple negative breast cancer, TNBC (0.714–1.43 mg/kg/day). Otherwise, Nguyen et al. [44] concluded through another in vivo study that morphine administration in a similar dose range (0.5–1.5 mg/kg/day) does not influence cancer initiation but promotes the progression of the existing tumour and cancer-related pain by mast cell activation and degranulation resulting in the shortened survival of C3TAG mice. Alternatively, Doorneball et al. [42] argued that in vivo, analgesic doses of morphine (10 and 20 mg/kg) do not affect breast tumour growth, angiogenesis, nor facilitate dissemination of metastases after surgery.

Taken together, these results suggest that morphine’s effects may alter tumour response in accordance with the duration of opioid exposure and the dosage used, which could influence the treatment of BC patients.

### 3.5. Limitations of This Study

There are several limitations in this review that preclude drawing definitive conclusions regarding the influence of analgesics and LAs on cancer progression and metastasis formation. This systematic review summarizes recent studies on the effect of lidocaine, ropivacaine, levobupivacaine and morphine on BC cell lines and animal models and in combination with other treatment regimens.

Firstly, heterogeneity between experimental designs was noted in the identified studies. Different cancer cell types may function distinctly, and as the overall understanding of malignancy evolves, it becomes important that the biochemical similarities and differences between cancer cell types are elucidated. Furthermore, different mechanisms of action were proposed for the same effect of LAs in similar BC cells, and as a result, no clear consensus was established. Secondly, the animal models included in this review reflect different aspects of the BC, and no animal model is a perfect match to represent the full clinical situation. Thus, the significance of these conclusions needs to be further examined to translate these findings in terms of clinical significance and establish the implications upon the outcome and prognosis of BC patients’ treatment.

Thirdly, in general, the concentration and duration of LAs’ isolated application required to exert a significant anti-tumour effect are not achievable through local infiltration. However, despite LAs bearing potential as synergistic agents of conventional therapies, there are few results regarding the effect of LAs and opioids in combination with different therapeutic approaches and there is a high variation of results between studies. In fact, studies that concern the effects that LAs may have in combination with chemotherapeutic agents, as well as their specificity to tumour cells are scarce.

In addition, concerning ropivacaine and levobupivacaine few in vitro studies were found and only one in vitro and in vivo study was found for ropivacaine. No in vivo studies were found for levobupivacaine. This influences the reliability of the conclusions drawn from this systematic review and arises the great need for more consistent reporting of essential details regarding experimental design for future animal experiments and studies of combination with other therapies.

## 4. Materials and Methods

The methods of this systematic review were performed following the PRISMA (Preferred Reported Items for Systematic Review and Meta-analysis) guidelines [27].

### 4.1. Search Strategy

This study is a systematic review to identify eligible preclinical studies concerning the use of pain medications and its influence in BC treatment response. A search on the literature was performed in Medline retrieved from PubMed, Scopus and Web of Science databases, using the searching formulas presented in Table 3. The filters of English language and publication date since 2010 were applied, considering that in 2010 a new adaptation of the World Health Organization (WHO) guideline for the treatment of cancer pain, “three-step analgesic ladder” promoted its bidirectional use with a “step up, step down” approach [49,50].

### 4.2. Inclusion and Exclusion Criteria

In order to be included in the analysis, the articles need to meet the following inclusion criteria: (i) preclinical studies, i.e., in vitro and/or in vivo; (ii) studies devoted to the study of ropivacaine, levobupivacaine, lidocaine, morphine or methadone; (iii) studies on BC field. Clinical studies, all types of reviews, editorials, opinion articles, preclinical studies non-related to the topic, articles written in other languages than English, written before 2010 and unavailable as complete articles were excluded. The search on the literature was undertaken in December 2021.

### 4.3. Data Collection, Extraction and Analyses

The articles found with the search were downloaded into the reference manager software (Mendeley Desktop v1.19.4) and duplicates were eliminated. Then, a first screening was performed based on the title and abstract by two independent reviewers. The selected articles were read in full for evaluating whether they meet all the inclusion and exclusion criteria. In case of non-consensus, a third reviewer was included in the selection process. For each eligible reference, descriptive and quantitative data were collected by the reviewers. In particular, the reviewers extracted data regarding to: (i) the authors and year of the publication; (ii) the type of study, i.e., in vitro or in vivo; (iii) the specific study characteristics as population, i.e., the cell line/animal model used, and the anaesthetic/analgesic used; (iv) the main outcomes, such as the range of concentration and the effects observed; (v) the limitations of the studies. The study research methodology is illustrated in Figure 1.

## 5. Conclusions

Local anaesthetics lidocaine, ropivacaine, and levobupivacaine present promising and consistent results regarding the anticancer influence of LA on BC cell lines, when administered alone or in combination with other treatment regimens. Thereby, this may positively affect patient outcomes and global survival. Although scarcely studied, levobupivacaine appears to be the most effective LA against BC studied in vitro. Nonetheless, overall results suggest that the LAs’ effects seem to be promoted in a selectively and dose-dependent manner, at concentrations clinically relevant (0.3 to 10 mM). Still, at lower concentrations, these LAs exhibit some effects on epigenetic information and cell cycle. In contrast, the use of morphine, while effective in the management of cancer-related pain, mostly seems to act as a tumour-promoting agent, despite showing anti-proliferative properties in combination with chemotherapeutic agents. However, overall results propose that morphine’s effects might change tumour response according to the dose and duration of opioid exposure, which could influence the treatment and clinical outcome of BC patients. In turn, there is a paucity of data regarding the influence of methadone on the therapeutic response to BC thus raising the need for future pre-clinical studies.

This review additionally highlighted the relevance of deepening the study of the mechanisms behind these controversial effects, in order to evaluate the benefits and disadvantages of using each LA or analgesics in specific clinical settings and diseases, allowing the optimization of its application for BC patients improving their cancer treatment outcome and prognosis.

## Figures and Tables

**Figure 2 ijms-23-01894-f002:**
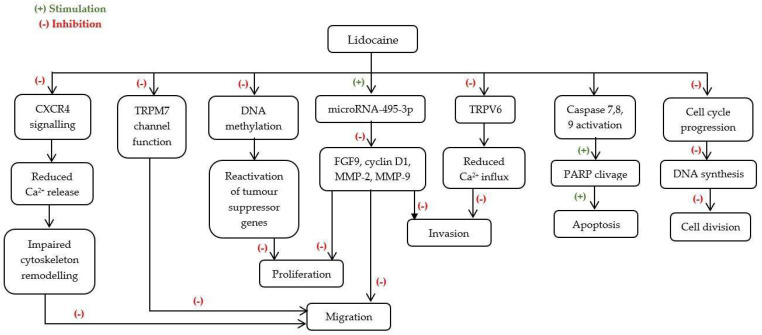
Possible mechanisms underlying lidocaine’s anti-cancer effects. Ca^2+^—Calcium; CXCR4—C-X-C Chemokine Receptor type 4; DNA—Deoxyribonucleic acid; FGF9—Fibroblast Growth Factor 9; MMP-2—Matrix Metallopeptidase 2; MMP-9—Matrix Metallopeptidase 9; PARP—Poly (ADP-ribose) Polymerase; TRPM7—Transient receptor potential cation channel subfamily M member 7; TRPV6—Transient receptor potential cation channel subfamily V member 6.

**Figure 3 ijms-23-01894-f003:**
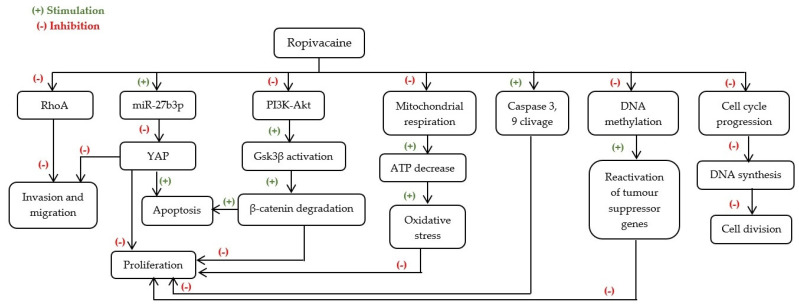
Possible mechanisms underlying ropivacaine’s anti-cancer effects. ATP—Adenosine triphosphate; DNA—Deoxyribonucleic acid; Gsk3β—Glycogen synthase kinase 3 beta; miR-27b3p—microRNA-27b3p; PI3K-Akt—Phosphatidylinositol 3-kinase—Protein kinase B; RhoA—Ras homolog family member A; YAP—Yes-associated protein.

**Figure 4 ijms-23-01894-f004:**
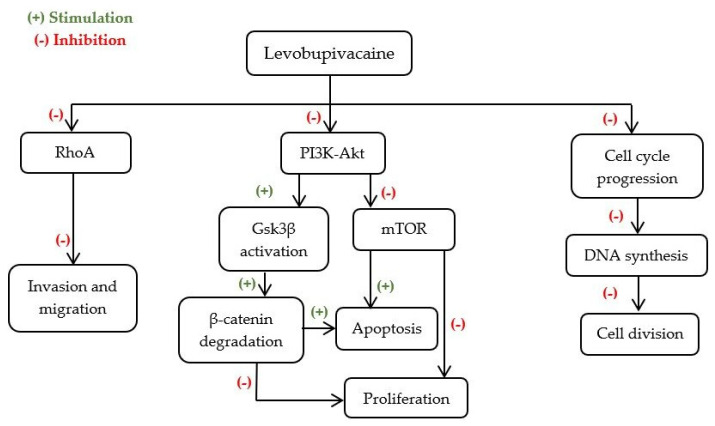
Possible mechanisms underlying levobupivacaine’s anti-cancer effects. DNA—Deoxyribonucleic acid; Gsk3β—Glycogen synthase kinase 3 beta; mTOR—Mammalian target of rapamycin; PI3K-Akt—Phosphatidylinositol 3-kinase—Protein kinase B; RhoA—Ras homolog family member A.

**Figure 5 ijms-23-01894-f005:**
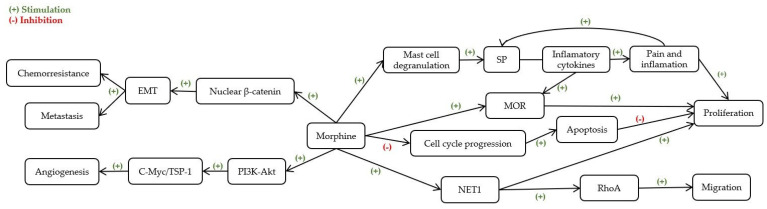
Possible mechanisms underlying morphine’s effects on breast cancer cells. EMT—Epithelial-mesenchymal transition; MOR—μ-opioid receptors; NET-1—Neuroepithelial Cell Transforming 1; PI3K-Akt—Phosphatidylinositol 3-kinase—Protein kinase B; RhoA—Ras homolog family member A; SP—Substance P; TSP-1—Thrombospondin-1.

**Table 1 ijms-23-01894-t001:** Clinically recommended concentrations of anaesthetics/opioids used in this systematic review.

Anaesthetic/Opioid	Injectable Concentration	Recommended Infusion Dose	Maximum Single Dose	Reference
Lidocaine	5–20 mg/mL	1.2 mg/kg/h	4.5 mg/kg (300 mg)	[19,20]
Morphine	2–10 mg/mL	0.1–0.2 mg/kg/4 h	No maximum dose	[21,22]
Ropivacaine	2.5–7.5 mg/mL	-	3 mg/kg (200 mg)	[19,20]
Levobupivacaine	2–10 mg/mL	-	2 mg/kg (150 mg)	[19,20]

**Table 3 ijms-23-01894-t003:** Search Strategy used.

Database	Search Formula
Medline (via PubMed)	((((“breast cancer”[TIAB]) OR “Breast Neoplasms”[TIAB] OR “Breast Neoplasms”[MESH]) OR “breast carcinoma”[TIAB])) AND (((((ropivacaine[TIAB]) OR ropivacaine[MESH] OR levobupivacaine[TIAB] OR levobupivacaine[MESH]) OR morphine[TIAB] OR morphine[MESH]) OR lidocaine[TIAB] OR lidocaine[MESH]) OR methadone[TIAB] OR methadone[MESH])
Scopus	TITLE-ABS-KEY((“breast cancer” OR “Breast Neoplasms” OR “breast carcinoma”) AND (ropivacaine OR levobupivacaine OR morphine OR lidocaine OR methadone))
Web of Science	(TS = ((“breast cancer” OR “Breast Neoplasms” OR “breast carcinoma”) AND “ropivacaine” OR “levobupivacaine” OR “morphine” OR “lidocaine” OR “methadone”)))

## Data Availability

Not applicable.

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
