# Peer review of "The Potential Effect of Lidocaine, Ropivacaine, Levobupivacaine and Morphine on Breast Cancer Pre-Clinical Models: A Systematic Review"

_ijms, 2022, doi:10.3390/ijms23031894_

Round 1

Reviewer 1 Report

The present review on the effects of 3 local anaesthetics and 1 opioid on breast cancer progression and metastasis formation.

The results are discussed according to the different cell lines used and their genetic differences and responses.

Did the Authors also searched for the commercial names of these commounly used drugs. This is not indicated in the Materials and Methods section, nor clear in the Workflow for search strategy presented in Table 3?

Problem with Figures' citations in the text is observed:

Line 104: Figure 1 (not "Error! Reference source not found. Error! Reference source not found..")

Line 147, 429, 455: same remark (not "Error! Reference source not found. Error! Reference source not found..")

Line 397: limitations

Author Response

We thank the Reviewers for their learned and critical comments on our paper. We have thoroughly pondered over all the questions raised and revised our manuscript in accordance with the Reviewers’ concerns. Below you can find the specific answers to the remarks of the Reviewer 1 written in blue.

Did the Authors also searched for the commercial names of these commounly used drugs. This is not indicated in the Materials and Methods section, nor clear in the Workflow for search strategy presented in Table 3?

Answer: We that the Reviewer pertinent question. In fact, we did not search by the commercial names of these commonly used drugs. However, we repeated the search using these commercial names and we found that such a search was not necessary as the same number of studies were retrieved and that in fact the retrieved studies also contained the name of the active substance, so the search only by the terms of the active substances was sufficient to retrieve the studies eligible for this analysis.

Problem with Figures' citations in the text is observed:

Line 104: Figure 1 (not "Error! Reference source not found. Error! Reference source not found..")

Answer: The Figure citation on line 104 has been changed.

Line 147, 429, 455: same remark (not "Error! Reference source not found. Error! Reference source not found..")

Answer: The Figures’ citations on lines 147 and 455 have been changed. The table citation on line 429 has been changed.

Line 397: limitations

Answer: The typo has been corrected.

Reviewer 2 Report

  1. Why do authors target only breast cancer in this study? Please justify.
  2. What is the anti-cancer mechanism of local anesthetics? Authors should include a representative figure for this. There is a number of reports available.
  3. What are the pros and cons of using local anesthetics as anticancer agents? Please discuss.
  4. There is a number of reports available about anticancer effects of local anesthetics but still, there are not developed as anti-cancer therapeutics, why? Please discuss.
  5. Please include a table about clinical trials of local anesthetics if there are some already under investigation.

Author Response

We thank the Reviewers for their learned and critical comments on our paper. We have thoroughly pondered over all the questions raised and revised our manuscript in accordance with the Reviewers’ concerns. Below you can find the specific answers to the remarks of the Reviewer 2 written in blue. 

Why do authors target only breast cancer in this study? Please justify.

Answer: This very important topic was addressed in more detail in the introduction and discussion.

According to the American Cancer Society, in 2020, both in Europe and worldwide, breast cancer (BC) was considered the most diagnosed type of cancer and the leading cause of mortality in women. Deaths from this pathology are commonly caused by metastization and tumour recurrence.

Early diagnosis and adequate treatment are associated with a greater probability of cure and a better prognosis. In fact, the early diagnosis tests commonly used are the clinical breast examination, imaging techniques (mammography, ultrasonography and magnetic resonance imaging) and in suspected cases, biopsy under local anesthesia. Concerning treatment, surgery, radiotherapy and systemic treatment, such as chemotherapy, are the three main treatments used in the therapeutic scheme for BC, with surgical resection of the primary tumour being the method associated with a better prognosis. Nevertheless, the perioperative period, due to perioperative factors such as pain, stress, and anesthetic techniques, seems to potential influence the development of metastases and tumour recurrence, which as indicated above, is associated with the mortality rate.

Moreover, these patients usually have cancer-related pain, especially in more advanced stages of disease or associated with treatment techniques.

To emphasize these ideas, some clarification statements were added in the proper sections, namely introduction (lines 37-42) and/or discussion (line 142). 1 more reference was added in introduction (line 42).

What is the anti-cancer mechanism of local anesthetics? Authors should include a representative figure for this. There is a number of reports available.

Answer: In vitro and in vivo studies have shown that LAs may reduce cancer recurrence and development by inhibiting cell proliferation, viability, invasion and migration and by promoting tumor cell death. However, the etiology of these effects is probably multifactorial, therefore we fully agree with the reviewer that we must provide more information. For that, we included representative figures in the section “3. Discussion”, namely in sections “3.1. Lidocaine”, “3.2. Ropivacaine”, “3.3. Levobupivacaine”, “3.4. Morphine”.

What are the pros and cons of using local anesthetics as anticancer agents? Please discuss.

Answer: In order to be as strict as possible, section “3. Discussion” section “3.5. Limitations” was rectified to better explain and identify in the manuscript the pros and cons of using local anesthetics as anticancer agents (lines 435-437).

Local anaesthesics (LAs) are commonly used as co-adjuvants on pain control therapy and in some technical procedures of diagnosis and treatment. These drugs have the advantage of providing effective analgesia and anesthesia, minimizing the adverse and immunosuppressive effects of pre- and postoperative pain and therefore reducing the required dose of analgesics (e.g. opioids) in the postoperative period. When used as a co-adjuvant they also allow reducing the dose of prescribed opioid, which translates into increased safety of therapy in terms of opioid-induced immunosuppression and lower potential risk of cancer progression. Alternatively, LAs are unlikely to be used as anti-tumor therapies alone as the concentration and duration of LAs application required to exert a significant anti-tumor effect are not achievable through local infiltration.

There is a number of reports available about anticancer effects of local anesthetics but still, there are not developed as anti-cancer therapeutics, why? Please discuss.

Answer: LAs are unlikely to be used as in anti-tumor therapies alone as the concentration and duration of LAs isolated application required to exert a significant anti-tumor effect are not achievable through local infiltration. However, studies suggest that LAs can be used as synergistic agents of conventional therapies such as chemotherapy and radiotherapy but the studies concerning the effect that LAs may have on the toxicity of chemotherapeutic agents as well as the sensitivity of tumor cells to them is scarce, which does not allow to draw a definitive conclusion about this ability of LAs, thus arises the need to be explored in future studies. This very interesting point that we missed in the manuscript was now added (lines 435-441).

Please include a table about clinical trials of local anesthetics if there are some already under investigation.

Answer: We thank the reviewer this pertinent question. Our main objective was to demonstrate the influence of morphine and local anaesthesics on breast cancer treatment response, so that in future our actions may help to overcome the limitations found in the studies analyzed, both in terms of the mechanism of the drugs alone and/or in combination with other anti-cancer agents. Therefore, our article focuses only on preclinical studies. Nevertheless, in order to be as strict as possible, the title has been changed as indicated at the beginning of the document.

Reviewer 3 Report

In this work, the Authors collected and analyzed the preclinical studies existing in the literature to assess the influence of the selected pain medications (lidocaine, ropivacaine, levobupivacaine, morphine) on breast cancer treatment response. 21 preclinical studies (both in vivo and in vitro, related to BC cell lines and animal models) were included in this valuable overview. The combination and effect of these drugs with anti-cancer chemotherapeutic agents was also discussed.

I consider the strengths of the work to be: an interesting topic that significantly contributes to the topic of breast cancer treatment, a rich range of literature (52 references, including web pages, most of the papers published after 2010 up to September 2021). In general, the manuscript is well organized, well described, conclusions drawn are in agreement with experimental data.

I only have two minor remarks:

  • Lines 104, 147, 429, 455: There are some errors here. Enter the appropriate reference numbers.
  • Line 397: Correct the typo in ‘imitations’.

Therefore, the recommendation of the manuscript for publication in its current form can be considered justified.

Author Response

We thank the Reviewers for their learned and critical comments on our paper. We have thoroughly pondered over all the questions raised and revised our manuscript in accordance with the Reviewers’ concerns. Below you can find the specific answers to the remarks of the Reviewer 3 written in blue.

I only have two minor remarks:

Lines 104, 147, 429, 455: There are some errors here. Enter the appropriate reference numbers.

Answer: The Figures’ citations on lines 104, 147, 455 have been changed. The table citation on line 429 has been changed.

Line 397: Correct the typo in ‘imitations’.

Answer: The typo has been corrected.